# Rapid Mosaicking of Unmanned Aerial Vehicle (UAV) Images for Crop Growth Monitoring Using the SIFT Algorithm

**Jianqing Zhao** [†] , **Xiaohu Zhang** [†] , **Chenxi Gao, Xiaolei Qiu, Yongchao Tian, Yan Zhu** and **Weixing Cao** *

National Engineering and Technology Center for Information Agriculture, Key Laboratory for Crop System Analysis and Decision Making, Ministry of Agriculture and Rural Affairs, PRC, Jiangsu Key Laboratory for Information Agriculture, Jiangsu Collaborative Innovation Center for Modern Crop Production, Nanjing Agricultural University,1 Weigang Road Nanjing, Jiangsu 210095, China; jianqingz@hotmail.com (J.Z.); zhangxiaohu@njau.edu.cn (X.Z.); gaochenxi_@hotmail.com (C.G.); qiuxiaolei@njau.edu.cn (X.Q.); yctian@njau.edu.cn (Y.T.); yanzhu@njau.edu.cn (Y.Z.)
* Correspondences: caow@njau.edu.cn
† Jianqing Zhao and Xiaohu Zhang have contributed to the work equally.

**Abstract:** To improve the efficiency and effectiveness of mosaicking unmanned aerial vehicle (UAV) images, we propose in this paper a rapid mosaicking method based on scale-invariant feature transform (SIFT) for mosaicking UAV images used for crop growth monitoring. The proposed method dynamically sets the appropriate contrast threshold in the difference of Gaussian (DOG) scale-space according to the contrast characteristics of UAV images used for crop growth monitoring. Therefore, this method adjusts and optimizes the number of matched feature point pairs in UAV images and increases the mosaicking efficiency. Meanwhile, based on the relative location relationship of UAV images used for crop growth monitoring, the random sample consensus (RANSAC) algorithm is integrated to eliminate the influence of mismatched point pairs in UAV images on mosaicking and to keep the accuracy and quality of mosaicking. Mosaicking experiments were conducted by setting three types of UAV images in crop growth monitoring: visible, near-infrared, and thermal infrared. The results indicate that compared to the standard SIFT algorithm and frequently used commercial mosaicking software, the method proposed here significantly improves the applicability, efficiency, and accuracy of mosaicking UAV images in crop growth monitoring. In comparison with image mosaicking based on the standard SIFT algorithm, the time efficiency of the proposed method is higher by 30%, and its structural similarity index of mosaicking accuracy is about 0.9. Meanwhile, the approach successfully mosaics low-resolution UAV images used for crop growth monitoring and improves the applicability of the SIFT algorithm, providing a technical reference for UAV application used for crop growth and phenotypic monitoring.

**Keywords:** Unmanned aerial vehicle; crop growth monitoring; image mosaicking; feature matching; SIFT

## 1. Introduction

Remote sensing from an unmanned aerial vehicle (UAV) is an emerging monitoring technique. Due to its advantages of convenience, high efficiency, and ability to fly at low altitudes under clouds, remote sensing from UAVs has been widely applied in various fields [1]. In the field of precision agriculture, the acquisition of high-resolution images through UAVs carrying different sensors is

an important approach for crop growth monitoring [2,3] and identification of pests, diseases, and weeds [4,5], and it exhibits an enormous potential in the field of crop phenotype monitoring [6,7].

At present, farmland crop growth is mainly monitored by rotary-wing and fixed-wing UAVs with UAV flight plans including flight altitude and flight speed [8,9]. These UAVs carry different sensors such as RGB, NIR, thermal infrared, and hyperspectral sensors, and images acquired by different sensors usually have different resolutions [2–4]. During the UAV flight, the location of camera is constantly changing and images have varying view angles [10]. The UAV image acquisition modes are mainly manual acquisition mode, fixed-point acquisition mode, and cruise acquisition mode [10]. In the manual mode for image acquisition, the operator of the UAV triggers the camera and takes photos manually during the flight. In the fixed-point mode, the UAV flies following a predefined path and stops at its location for image acquisition. In the cruising mode, the images are taken during the flight without stopping flying.

All three modes above can steadily capture and obtain images over the current field coverage, but for a large area, an appropriate image mosaicking technique is needed to mosaic farmland UAV images over the area to form a complete image of the entire target region [11]. Mosaicking of UAV images refers to the technique that merges an obtained sequence of two or more UAV images into one wide-field image, and the main procedures include image preprocessing, image registration, and image fusion [10,12]. Image registration establishes the transformation model relationship between image features, achieves image matching through scale-space and is the core procedure of image mosaicking [12]. As the most important procedure in the mosaicking of UAV images, the registration accuracy directly affects the quality of UAV images in crop growth monitoring.

Commercial software and open-source image mosaicking techniques are the main tools for mosaicking UAV images used for crop growth monitoring. Commercial aerial image mosaicking software, such as Agisoft® Photoscan (http://www.agisoft.com), Pix4D® Pix4Dmapper (https://www.pix4d.com), and Blue Marble Geographics® Global Mapper (http://www.bluemarblegeo.com), can be used to mosaic UAV images for crop growth monitoring. Such software supports a graphical interface and is easy to operate and master. These commercial software programs are oriented to photogrammetry and pay attention to the spatial accuracy of registration. Photoscan and Pix4D generate orthomosaics based on orthorectification. They remove the perspective distortions from the images using the DSM that is generated from the 3D Model and export orthomosaic as a basis for professional quality GIS data. Different from Photoscan and PIX4D, multiple images can be mosaicked or stitched together to form one contiguous file for Global mapper without 3D reconstruction. However, the computational efficiency of commercial software is relatively low [11], and when agricultural researchers want to monitor real-time crop growth in the field, they do not need to use highly accurate orthomosaics. In addition, for the purpose of real-time crop growth monitoring, the open-source method is a better choice to integrate image mosaic function in a real-time UAV monitoring system [13]. According to the image information matching method, open-source mosaicking techniques are classified as either transformation domain-based image registration, gray value-based image registration, or feature-based image registration [14]. In particular, feature-based image matching is the mainstream technical method for mosaicking of farmland UAV images. Feature-based matching includes point matching, line matching, and area matching, and can manually or automatically detect salient and unique objects [15]. In particular, when this process is used for the matching of feature points, the feature points are also called the control points [15,16]. By extracting the common features between two images and using a feature-based matching algorithm, we can reduce the influence of noise in the feature extraction process, which makes this method suitable for the matching of remote sensing images [17].

As a matching method based on features, the algorithm called scale-invariant feature transform (SIFT) was proposed by David G. Lowe of the University of British Columbia in 1999 and was improved and perfected in 2004 [18,19]. SIFT exhibits invariance to image translation, rotation, and scaling transformations and good robustness to light changes, noise, and affine transformation [19,20]. SIFT is the main extraction method for feature points and has been applied to the matching of agricultural

remote sensing visible and multispectral images, crop classification, and pest identification, and all these applications achieve good results [4,21,22].

SIFT is one of the most widely used and successful image processing methods from the past 15 years. Previous studies have improved SIFT from two perspectives: the feature descriptor extraction and the feature point matching. To improve the feature descriptor extraction algorithm, principal component analysis (PCA) and independent component analysis (ICA) have been used to reduce the dimensions of the feature descriptors, therefore decreasing computation requirements and increasing the efficiency of the SIFT algorithm [23,24]. Meanwhile, the gradient location-orientation histogram (GLOH) and uniform robust SIFT (UR-SIFT) have been used to improve the computation of gradient histograms of feature points to improve the robustness of the SIFT algorithm [25,26]. To improve the feature point matching algorithm, the Lissajous curve-based similarity measurement and the multiscale Harris corner detector have been used to eliminate a large number of mismatched points to extract the feature points with high reliability [27,28]. However, the above studies were aimed at improving the general SIFT algorithm itself. As researchers attempt to monitor real-time crop growth [29], a fast mosaicking algorithm accounting for UAV images with prior knowledge of the image is needed.

By combining the previous studies and the characteristics of UAV operation for crop growth monitoring, we found that the overlap, texture features, and structure content of crop images from farmlands affect the efficiency and accuracy of mosaicking. Crop growth monitoring by using UAVs is usually performed at low flight altitudes, and the images have a high degree of overlap [30,31]. For thermal infrared images, the images have low resolution due to the limitations of the sensors, and the texture features of crop images are not obvious [32]. These factors profoundly affect the number of matched feature points. Meanwhile, because the UAV images in crop growth monitoring contain many artificial farmland structures (such as ditches and roads) with high degrees of similarity, the SIFT algorithm generates many mismatched point pairs, causing a reduction in mosaicking accuracy or mosaicking failure.

To meet the needs of real-time crop growth monitoring, a fast method for mosaicking UAV images based on SIFT is proposed in this paper. This method can enhance the efficiency and applicability of mosaicking, and the result can be easily browsed and shared on mobile terminals. This method provides technical reference and software tools for large-scale application of UAV images in crop growth monitoring.

The paper is organized as follows. First, Section 1 presents related work on UAV images acquisition modes and mosaicking methods including commercial software and open-source methods. Section 2 explains the rapid UAV image-mosaicking method using the SIFT algorithm for crop growth monitoring. Section 3 presents the experiment and the comparison method. Section 4 presents the results of the proposed method, and discusses the efficiency and the accuracy. Finally, Section 5 presents the conclusions and direction of future work.

## 2. Methodology

### 2.1. Image Mosaicking Method Based on the Standard SIFT Algorithm

Image mosaicking based on the standard SIFT (SSIFT) algorithm uses the SIFT algorithm to extract and match the feature points in the images to be merged, establish the matched feature point pair sets, and simultaneously inspect and delete the mismatched point pairs. We use the matched feature point pair sets to establish a transformation equation in the image space and therefore register and fuse the images to be merged [28] (Figure 1).

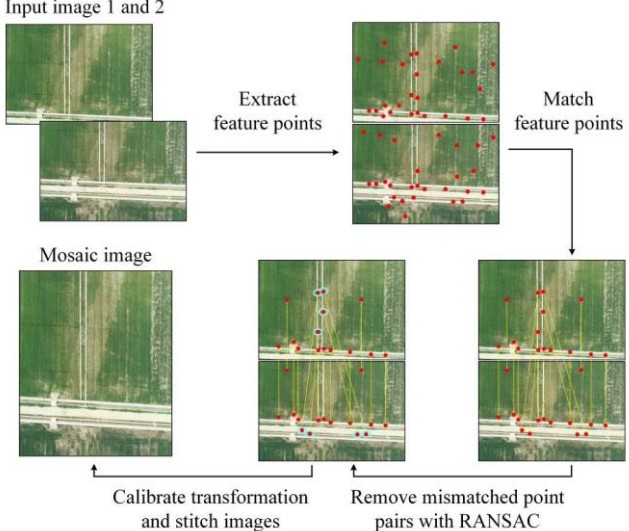

**Figure 1.** Flowchart of standard scale-invariant feature transform (SSIFT)-based image mosaicking.

### 2.1.1. Extraction of Feature Points from Images

The SIFT algorithm uses the difference of Gaussians to establish the difference of Gaussian (DOG) scale-space. On images with different spatial scales, the feature points are extracted, and the unstable feature points are eliminated; the principal direction of the feature points is determined, and the SIFT feature descriptor is generated, avoiding the mismatch caused by rotation, illumination, scale, and noise [18].

The basic idea of scale-space is to obtain visual processing information under different scales and determine the essential features of these images through comprehensive analyses [18]. By combining with the Gaussian function $G(x, y, \sigma)$ of scale factor $\sigma$, we can establish scale-space $L(x, y, \sigma)$ of image $I(x, y)$, which is abbreviated as $L$. To improve the efficiency of feature point extraction, through scale-space $L(x, y, \sigma)$, SIFT further constructs the DOG scale-space $D(x, y, \sigma)$, which is abbreviated as $D$. The detailed process of construction process is given by Lowe [19].

In the DOG scale-space $D$ of image $I(x, y)$, every pixel of every image is compared with eight adjacent pixels at the same scale and nine adjacent pixels of images at the two nearest scales. If the value is minimum or maximum at this pixel, it is considered as a feature point. However, the positions of feature points obtained in this way are offset, and we need to delete the feature points with low contrast [19]. Thus, we conduct the Taylor series expansion of scale-space function $D$ at point $A$ (Equation (1)) to obtain a more accurate position of feature point $A'$.

$$D = D_A + \frac{\partial D_A^T}{\partial \mathbf{x}}\mathbf{x} + \frac{1}{2}\mathbf{x}^T\frac{\partial^2 D_A}{\partial \mathbf{x}^2}\mathbf{x} \tag{1}$$

where $\mathbf{x} = (x, y, \sigma)^T$ is the offset from point $A$ to point $A'$. Lowe et al. calculated the derivative on both sides of $D(\mathbf{x})$ to obtain the precise location $\hat{\mathbf{x}}$ of feature point $A'$ [19] (Equation (2)) and also to compute the DOG scale-space function $D(\hat{\mathbf{x}})$ of $A'$ (Equation (3)).

$$\hat{\mathbf{x}} = -\frac{\partial^2 D_A^{-1}}{\partial \mathbf{x}^2}\frac{\partial D_A}{\partial \mathbf{x}} \tag{2}$$

$$D(\hat{\mathbf{x}}) = D_A + \frac{1}{2}\frac{\partial D_A^T}{\partial \mathbf{x}}\hat{\mathbf{x}} \tag{3}$$

$D_c$ is set to the threshold of contrast $|D(\hat{\mathbf{x}})|$ in the DOG scale-space and is used to measure the stability of feature points [19]. $D_c$ is usually set to 0.03, and all feature points with $D_c$ smaller than 0.03 are deleted [23]. However, many researchers note that the default $D_c$ is not suitable for all images,

causing low efficiency and accuracy for the extraction of feature points [33]. Therefore, for different types of UAV images used for crop growth monitoring, we need to select the appropriate contrast threshold in the DOG scale-space.

### 2.1.2. Removal of Mismatched Points

The direct application of the feature point set extracted by the SIFT algorithm in mosaic produces a large number of mismatched point pairs, which causes a reduction in mosaicking accuracy or even failure of mosaicking [34]. The random sample consensus (RANSAC) algorithm has strong ability to improve the matching accuracy of feature point pairs [35]. The RANSAC algorithm was proposed in 1981 by Fischler and Bolles and can eliminate the anomalous values from a set of observation data by iteratively calculating the parameters of mathematical models [36]. In particular, UAV images used for crop growth monitoring often has high degrees of similarity; the image features of many artificial farmland structures are extremely similar. If we want to obtain the ideal result, we need to iterate the calculation continuously to improve the parameters of the RANSAC algorithm, which increases the calculation costs [37,38]. Therefore, for crop growth monitoring using UAV images, it is necessary to combine a simple method to quickly remove mismatched feature point pairs.

### 2.2. Proposed Mosaicking Method Based on SIFT and RANSAC

For the aforementioned problem, in this paper, we propose a simple mosaicking method to increase the efficiency and accuracy of UAV image mosaicking in crop growth monitoring. Due to the characteristics of crop UAV images, the method optimizes the dynamic setting of the contrast threshold $D_c$ in the DOG scale-space in SIFT to improve the efficiency of the algorithm; we delete the mismatched point pairs to improve the mosaicking accuracy based on the relative position relationships of the UAV images. The details of the algorithm is shown in Algorithm 1.

---

**Algorithm 1** The procedure for the proposed algorithm.

---

**Input: A Set of UAV Images *I***
**Output: Mosaic Result *U***
1 Construct contrasts of image for set $C = \{C_i\}_{i=1}^N$ by Equation (4);
2 Compute average contrast $C_p$ of image by Equation (5);
3 **if** *I* are obtained by visible-light or near-infrared cameras **then**
4     Compute $D_{init}$ by Equation (6);
5 **end if**
6 **if** *I* are obtained by thermal infrared cameras **then**
7     Compute $D_{init}$ by Equation (7);
8 **end if**
9 Construct corresponding feature point sets *F* using $D_{init}$;
10 Initialize $k_1 = 1.1$, $k_2 = 1/1.1$, $P_{min}$, $P_{max}$, *F*;
11 **repeat**
12     Step:
13         Update $D_{new}$ by Equation (8);
14 **until** *F* satisfies;
15 Update *F* by Equations (9) and (10);
16 Initialize $P_{row}$, $P_{col}$;
17 **if** $I_p$ and $I_{p+1}$ have longitudinal overlap **then**
18     Update *F* using $P_{row}$ by Equations (11) and (12);
19 **end if**
20 **if** $I_p$ and $I_{p+1}$ have transverse overlap **then**
21     Update *F* using $P_{col}$ by Equations (13) and (14);
22 **end if**
23 *U* are acquired by *F*.

---

### 2.2.1. Determination of the Dynamic Contrast Threshold

The method uses the image contrast $C$, which can express the difference in image details and evaluate the characteristics of image quality [39], and this method is used to calculate the new contrast threshold $D_c$ in the DOG scale-space, which is denoted by $D_{init}$. To retain the local features of the UAV image, for a given set of UAV image sequences $I = \{I_p\}_{p=1}^{Q}$, we slide the $3 \times 3$ window on the UAV image $I_p$ to obtain the subdomain $Z = \{(I_p)_i\}_{i=1}^{N}$. We calculate the image contrast $C$ in each subdomain $Z_i$ and sort the $C$ values from small to large to obtain the sequence of image contrasts $C = \{C_i\}_{i=1}^{N}$ and the image contrast $C_p$ of UAV image $I_p$ (Equations (4) and (5)).

$$C = \left[ \frac{1}{N-1} \sum_{i=1}^{N} (x_i - \overline{x})^2 \right]^{\frac{1}{2}} \tag{4}$$

$$C_p = \frac{\sum\limits_{i=1}^{N} C_i}{N} \tag{5}$$

where $\overline{x}$ is the average gray value of pixel points in the subdomain and $x_i$ is the gray value of pixel $i$ in the subdomain. In the $3 \times 3$ subdomain, the number of pixel points $N$ is 9.

The textural features, illumination, and resolution in the visible, thermal infrared, and near-infrared crop images differ, and the image contrasts are also not the same [40,41]. The visible or near-infrared farmland crops images have high resolution, and the textural features are distinct. The number of feature points extracted by SSIFT is more than needed. When we are given a set of visible or near-infrared UAV images, the proposed method reduces the number of matched feature points between two images to improve the detection efficiency of feature points (Equation (6)).

$$D_{init} = D_c \times \frac{C_p}{C_{me} - C_p} \tag{6}$$

where $C_{me}$ is the median of the image contrast sequence $\{C_i\}_{i=1}^{N}$.

Thermal infrared images of farmland crops have the disadvantages of indistinct texture features and low resolution, which causes low contrast, and the number of feature points extracted by SSIFT is fewer than needed [42]. When we are given a set of thermal infrared UAV images, the proposed method increases the number of matched feature points between two images and ensures sufficient feature points for matching, increasing the matching accuracy (Equation (7))

$$D_{init} = D_c \times \frac{C_{me} - C_p}{C_p} \tag{7}$$

The proposed method searches for a more appropriate contrast threshold $D_{new}$ in the DOG scale-space on the basis of $D_{init}$. The method sets the minimum value $P_{\min}$ and maximum $P_{\max}$ for the number of feature points and unifies the number of feature points in a reasonable range. This step ensures that enough feature points are involved in matching and computing the transformation model. The method sets $k_1 = 1.1$ and $k_2 = 1/k_1$, iteratively calculates the number of matched feature points ($F$) between UAV images, and judges whether $F$ satisfies the restriction of the threshold to obtain the final contrast threshold $D_{new}$ in the DOG scale-space (Equation (8)).

$$D_{new} = \begin{cases} k_1 \times D_{init} & F > P_{\max} \\ D_{init} & P_{\min} < F < P_{\max} \\ k_2 \times D_{init} & F < P_{\min} \end{cases} \tag{8}$$

### 2.2.2. Improved Removal of Mismatched Points

An agricultural UAV flies according to a designed fixed route (Figure 2). The images have high degrees of overlap, and the spatial locations are relatively translated. The feature points to be matched should be located in the overlapping region. Therefore, we want to remove feature point pairs not in the overlap region and improve the matching accuracy with longitudinal overlap and transverse overlap.

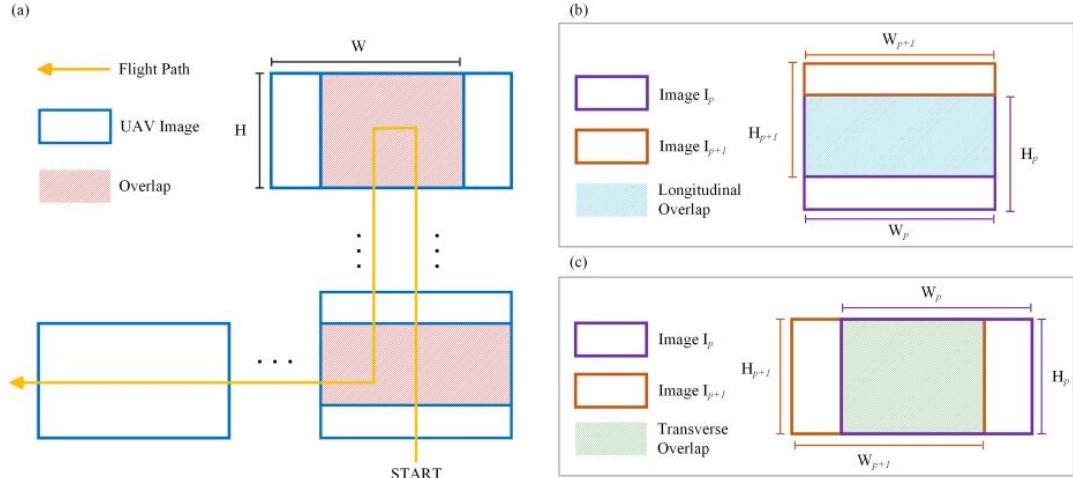

**Figure 2.** Schematic diagram for an unmanned aerial vehicle (UAV) capturing images in crop growth monitoring. In (**a**), the orange line represents the flight trajectory of the UAV; the blue rectangles represent the captured UAV images; and the shading in the blue rectangles represents the overlap of images. (**b**) represents the transverse overlap with 70% and (**c**) represents the longitudinal overlap with 75%.

We use a file coordinate system (row, col) which is defined in the image file. Its coordinate axes are parallel to the row and column directions of the digital image. Its units are pixels, and its origin is at the left upper corner of the left upper pixel [43]. Each feature point has its own row number and column number. The difference between the row number of each feature point pairs as well as the column number can be used to remove mismatched feature points. Therefore, the proposed method calculates the row number difference $\Delta\theta$ and column number difference $\Delta\varphi$ for every pair of matched feature points in reference image $I_p$ and image to be matched $I_{p+1}$, and combines this information with the overlap of $I_p$ and $I_{p+1}$ to set the constraint condition to delete the mismatched point pairs (Equations (9) and (10)).

$$H_{P+1} \times (1 - P_{row}) < |P_\theta - Q_\theta| < H_{P+1} \times P_{row} \tag{9}$$

$$W_{P+1} \times (1 - P_{col}) < \left|P_\varphi - Q_\varphi\right| < W_{P+1} \times P_{col} \tag{10}$$

where $P_\theta$, $Q_\theta$, and $P_\varphi$, $Q_\varphi$ represent the row number and column number where the matched feature points $P$ and $Q$ of two images $I_p$ and $I_{p+1}$ are located, respectively. $H_{P+1}$ and $W_{P+1}$ are the length and width of the image to be matched $I_{p+1}$, respectively. $P_{row}$ is the overlap rate of two images in the longitudinal direction, and $P_{col}$ is the overlap rate of two images in the transverse direction.

We consider two situations to delete feature point pairs not in the overlap region. When two adjacent images $I_p$ and $I_{p+1}$ have longitudinal overlap (Figure 2b), the proposed method uses the longitudinal overlap rate $P_{row}$ of $I_p$ and $I_{p+1}$ to eliminate the mismatched point pairs (Equations (11) and (12)), where $H_P$ and $W_P$ are the length and width of the image to be matched $I_p$, respectively.

$$H_P \times P_{row} < P_\theta < H_P \tag{11}$$

$$0 < Q_\theta < H_{P+1} \times P_{row} \tag{12}$$

When $I_p$ and $I_{p+1}$ have transverse overlap (Figure 2c), the proposed method uses the transverse overlap rate $P_{col}$ of $I_p$ and $I_{p+1}$ to delete the mismatched point pairs (Equations (13) and (14)).

$$W_P \times P_{col} < P_\varphi < W_P \qquad (13)$$

$$0 < Q_\varphi < W_{P+1} \times P_{col} \qquad (14)$$

By synthesizing the aforementioned improved algorithm, we can use the features of UAV images used for crop growth monitoring to find the appropriate $D_{new}$ and remove the improper feature point pairs to generate mosaic result $U$ and to improve the matching accuracy of feature points and reduce the image mosaicking time.

## 3. Experiment

### 3.1. Experiment Design

In this paper, we used a DJI® Jingwei M210 UAV to carry DJI® ZENMUSE X4S, DJI® ZENMUSE XT and MicaSense® RedEdge M cameras, which photograph from a height of 50 m, and are used to acquire a UAV image dataset for rice and wheat growth monitoring in Xinghua City, Jiangsu Province, China (119.90° E, 33.07° N) (Figure 3). This platform collects three types of crop growth monitoring data: visible (VI), near-infrared (NIR), and thermal infrared (TIR) images (Table 1). We set the longitudinal overlap as 75% and the transverse overlap as 70%. We also set the flight speed to 1 m/s and used fixed-point mode of image acquisition to obtain images clearly. The DJI UAV control system can automatically make a flight route according to region size, overlap rate, and other parameters.

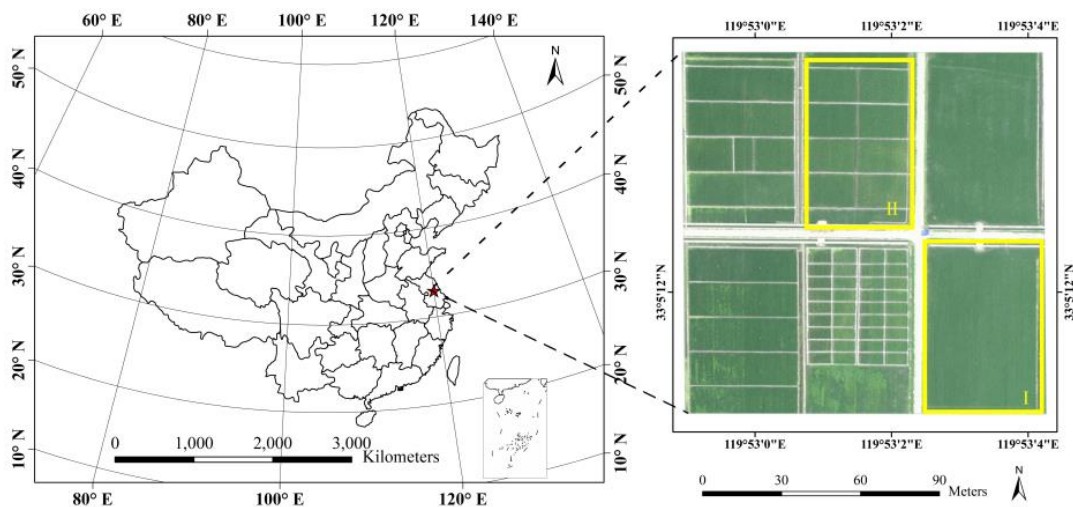

**Figure 3.** Region I and II of the research area in Xinghua, Jiangsu Province, China.

**Table 1.** UAV image data used in the experiment.

| Dataset | Category | Sensor | Resolution (pixels) | Phenology | Region | Area (ha) | Crop |
|---|---|---|---|---|---|---|---|
| 1 | VI | ZENMUSE X4S | 4864 × 3078 | Jointing | I | 0.36 | Wheat |
| 2 | VI | ZENMUSE X4S | 4864 × 3078 | Jointing | II | 0.38 | Rice |
| 3 | VI | ZENMUSE X4S | 4864 × 3078 | Maturity | II | 0.38 | Rice |
| 4 | TIR | ZENMUSE XT | 640 × 512 | Jointing | I | 0.36 | Wheat |
| 5 | TIR | ZENMUSE XT | 640 × 512 | Jointing | II | 0.38 | Rice |
| 6 | TIR | ZENMUSE XT | 640 × 512 | Maturity | II | 0.38 | Rice |
| 7 | NIR | RedEdge M | 1280 × 960 | Jointing | II | 0.38 | Rice |
| 8 | NIR | RedEdge M | 1280 × 960 | Maturity | II | 0.38 | Rice |

The image mosaicking based on SSIFT is compared to the method proposed in this study and the commercial mosaicking software Photoscan (PS) to evaluate the efficiency and accuracy of the three mosaicking methods. Photoscan has a simple interface and it enables the generation of sparse, dense point cloud, accurate three-dimensional textured meshes, and other products such as digital surface models and orthomosaics [44]. The longitudinal overlap and the transverse overlap we set is applicable for Photscan to process images correctly according to Agisoft PhotoScan User Manual. The basic configuration for the computer used in the image mosaicking experiment consists of an Intel® Core™ i7-7820X @3.60 GHz processor, Kingston® DDR4 2400 32G memory, an NVIDIA® GeForce GTX™ 1080 Ti graphics card, and the Microsoft® Windows™ 10 (64 bit) operating system. The proposed method conducts the functions in the Microsoft® Visual Studio 2017 environment.

*3.2. Comparison Method*

In this paper, we adopted mosaicking time consumption (T) and structural similarity index (SSIM) to evaluate the time efficiency and accuracy, respectively, of the proposed method. We focused on stitching images for real-time crop growth monitoring while the general whole workflow of UAV applications contains stitching, orthorectifying, and geo-referencing images [30]. Furthermore, the root-mean-square error (RMSE) is always used to evaluate the accuracy of geo-referencing, not the stitching. The accuracy of geo-referenced mosaics or orthomosaics can be evaluated by RMSE with ground control points (GCPs) [45,46], because the geo-locations of GCPs are measured and predicted. However, the geo-locations of GCPs in mosaics or orthomosaics without geo-referencing are not predicted, so we cannot apply RMSE as an accuracy evaluation index in our method. As a visual model-based indicator, SSIM is more in line with human visual characteristics [47] and measures the similarity of two static images from the perspectives of brightness, contrast, and structure [48]. Therefore, SSIM can evaluate image mosaicking more appropriately in this paper and is specifically defined as follows:

$$SSIM(x, y) = \frac{\left(2\mu_x\mu_y + C_1\right)\left(2\sigma_x\sigma_y + C_2\right)}{\left(\mu_x{}^2 + \mu_y{}^2 + C_1\right)\left(\sigma_x{}^2 + \sigma_y{}^2 + C_2\right)} \tag{15}$$

where $\mu_x$ and $\mu_y$ are the average brightness of two images, $\sigma_x$ and $\sigma_y$ are the standard deviations for the image gray values. $C_1$ and $C_2$ are two small constant correction factors. SSIM of mosaics can be obtained by calculating SSIM of overlapping regions with UAV images and taking average value. The range for SSIM values is [0, 1], and a large value indicates that the contents of two images tend to be more consistent. A value of 1 means that two images are completely identical, and a value of 0 represents two images with nonrelated content [48,49].

## 4. Results and Discussion

In this paper, we used SSIFT, the proposed method, and PS to mosaic UAV images for crop growth monitoring and compare the three mosaicking methods (Table 2). In comparison with the SSIFT-based image mosaicking method, the proposed method could increase the time efficiency of mosaicking by approximately 30% while this increase is approximately 20% compared with PS. All steps of workflow including alignment, calibration, dense point cloud, and reconstruction of three-dimensional image in Photoscan were set up at the "high" input. Some processing results of the Photoscan report are presented in Table 3. It is shown that the process of Photoscan extracts many feature points, which is one of the reasons why it is more time-consuming.

The proposed method can satisfy the need of agricultural researchers to monitor crop growth quickly and accurately with only output images in ordinary formats such as JPG. The proposed method does not need to output orthomosaics through three-dimensional reconstruction or other calculations, which is also one of the reasons why it has better time efficiency than PS. However, we did not use GPU hardware to accelerate the parallel computation, which could further optimize the performance. Because the GPU has stable and highly efficient computation capability, it is suitable for applications

with dense operation and high parallel computation requirements [50,51]. In future studies, we will add the acceleration technique based on the GPU to further improve the mosaicking efficiency of UAV images in crop growth monitoring.

**Table 2.** Results of mosaicking UAV images for the crop growth monitoring experiment.

| Dataset | Category | Method | Number of Images | T (s) | SSIM |
|---------|----------|--------|------------------|-------|------|
| 1 | VI | SSIFT | 8 | 2629 | 0.889 |
|   |    | Proposed method | 8 | 1719 | 0.913 |
|   |    | PS | 8 | 2060 | 0.873 |
| 2 | VI | SSIFT | 9 | 3146 | 0.909 |
|   |    | Proposed method | 9 | 1952 | 0.921 |
|   |    | PS | 9 | 2200 | 0.902 |
| 3 | VI | SSIFT | 9 | 3897 | 0.906 |
|   |    | Proposed method | 9 | 2060 | 0.915 |
|   |    | PS | 9 | 2330 | 0.901 |
| 4 | TIR | SSIFT | 12 | -* | - |
|   |    | Proposed method | 12 | 35 | 0.896 |
|   |    | PS | 12 | - | - |
| 5 | TIR | SSIFT | 10 | - | - |
|   |    | Proposed method | 10 | 19 | 0.898 |
|   |    | PS | 10 | - | - |
| 6 | TIR | SSIFT | 10 | - | - |
|   |    | Proposed method | 10 | 17 | 0.907 |
|   |    | PS | 10 | - | - |
| 7 | NIR | SSIFT | 9 | 64.92 | 0.834 |
|   |    | Proposed method | 9 | 45.18 | 0.887 |
|   |    | PS | 9 | 60 | 0.782 |
| 8 | NIR | SSIFT | 9 | 73.23 | 0.830 |
|   |    | Proposed method | 9 | 56.18 | 0.889 |
|   |    | PS | 9 | 63 | 0.812 |

* The hyphens in this table mean the failure of mosaicking.

**Table 3.** Results of Photoscan (high settings).

| Dataset | Number of Images | Number of Tie Points | Number of Extracted Points | Ground Resolution (cm/pixel) |
|---------|------------------|----------------------|----------------------------|------------------------------|
| 1 | 8 | 11,596 | 12,793 | 1.4 |
| 2 | 9 | 11,830 | 12,291 | 1.4 |
| 3 | 9 | 14,879 | 15,189 | 1.4 |
| 4 | 12 | 1402 | 1690 | 18.1 |
| 5 | 10 | 1377 | 1622 | 18.1 |
| 6 | 10 | 1763 | 1891 | 18.1 |
| 7 | 9 | 5457 | 6562 | 3.4 |
| 8 | 9 | 9556 | 10,261 | 3.4 |

On the other hand, the SSIM of the proposed method is approximately 0.9, which indicates that in comparison with the original image obtained by the UAV, the structure and content are essentially consistent and the mosaicking accuracy is high. Meanwhile, the SSIM values of the proposed method are 2.2–6.8% higher than those of the SSIFT method, so the proposed method can further increase the accuracy of mosaicking UAV images used for crop growth monitoring. PS automatically filters edge information from the UAV images to ensure the accuracy of orthomosaic, so SSIM is not very high. The proposed method contains the edge information well, and the mosaic image covers a wider range of

areas. In addition, the proposed method successfully conducts mosaicking on low-resolution thermal infrared crop images from farmland, which compensates for the limitations of the SSIFT-based and PS-based image mosaicking methods in failing to merge thermal infrared images and expands the applicability of the SIFT algorithm in mosaicking UAV images used for crop growth monitoring.

Meanwhile, the existing refined SIFT-based algorithm always focuses on generating and matching feature points more accurately [52–54]. In addition, due to the complex computation, the efficiency of these refined algorithms is reduced. Compared with these algorithms, we obtained a balance between efficiency and accuracy.

### 4.1. Dynamic Setting for the Contrast Threshold

The contrast threshold $D_c$ of the UAV images in DOG scale-space deeply affects the number of matched feature point pairs extracted from the UAV images used for crop growth monitoring [33,34]. The number of feature points rapidly decreases as $D_c$ decreases and eventually becomes stable (Figure 4a–c). When the factors of scene and sensor type are combined, no contrast threshold $D_c$ in DOG scale-space can be applicable to all the UAV images used for crop growth monitoring, but there is a reasonable range for the number of feature point pairs that allows relatively high efficiency and accuracy for matching these points [17]. When $D_c$ is 0.03, the number of feature point pairs extracted from the visible and near-infrared UAV images used for crop growth monitoring is more than needed while the number of feature point pairs extracted from thermal infrared images is fewer than needed, which seriously limits the applicability of the SIFT algorithm (Figure 4a–c).

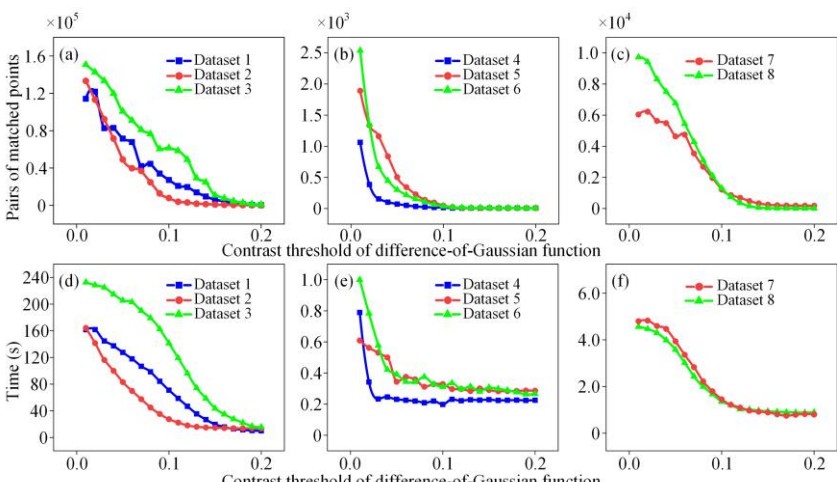

**Figure 4.** Number of matched feature points and matching time for the crop growth UAV images under different contrast thresholds in DOG scale-space. (**a**), (**b**), and (**c**) show the numbers of matched feature points for visible, thermal infrared, and near-infrared images, respectively. (**d**), (**e**), and (**f**) show the times for the matching of feature points for visible, thermal infrared, and near-infrared images, respectively.

The image contrast is one of the parameters of the gray level co-occurrence matrix that reflects the texture features of images [39] and is often used as the index to determine the contrast threshold in DOG scale-space [20]. The image contrast is not different in various types of UAV images used for crop growth monitoring, and all image types have applicable ranges for the number of feature point pairs. In these ranges, the numbers of feature point pairs are appropriate, the time consumption is relatively small, and the matching accuracy is not changed [20]. When the numbers of feature point pairs extracted from the UAV visible and near-infrared images used for crop growth monitoring are under 3000 and 2000, respectively, the influence of changing $D_c$ on the extraction of feature points is small (Figure 4a,c). When the number of feature point pairs extracted from thermal infrared images is less than 500, although the decreasing trend slows, the number rapidly approaches 0. At this time,

the matching accuracy of extracted feature points is low (Figure 4b). Therefore, in this paper, we set the appropriate range for the number of feature point pairs between 500 and 3000, which can essentially satisfy the registration requirements for the three types of UAV images used for crop growth monitoring.

### 4.2. Removal of Mismatched Point Pairs

The images obtained by UAVs at low altitudes have a high degree of overlap and high image resolution [15]. The UAV images in crop growth monitoring contain artificial farmland structures (such as ditches and roads) with a high degree of similarity. These factors generate many mismatched point pairs and affect the mosaic quality [55]. If we wish to further improve the matching accuracy, we need to constantly increase the number of RANSAC iterations, which will increase the computation time [56]. Therefore, based on RANSAC, this paper proposes a method to quickly reduce the impact of mismatching and the complexity of computation with the relative position relationships of UAV images.

If the feature points of two images are matched correctly, the difference in the row number and column number of the images where the feature point pairs are located lie within an appropriate threshold range. If the matches are incorrect, anomalies appear in the column numbers differences. The method based on relative position uses the information on the differences in the coordinates of feature points and can effectively eliminate mismatched point pairs caused by the artificial structures of roads and ditches to improve the accuracy of imaging mosaicking. The proposed method can delete the feature point pairs with excessively large and excessively small differences in row and column numbers and unify the differences in row and column numbers of feature point pairs within an appropriate threshold range. In comparison with using only the RANSAC algorithm, we further delete essentially more mismatched feature point pairs than the number of correct matches (Figure 5) and significantly decrease the number of feature point pairs that finally participate in the registration (Figure 6). The proposed method has an even higher SSIM, which proves that the method of deleting mismatched point pairs based on their relative positions in images can effectively improve the accuracy of image mosaicking and guarantee the accuracy of visual observation. Moreover, the calculation of the proposed method is concise, which avoids any reduction in the time efficiency of mosaicking due to additional computation requirements.

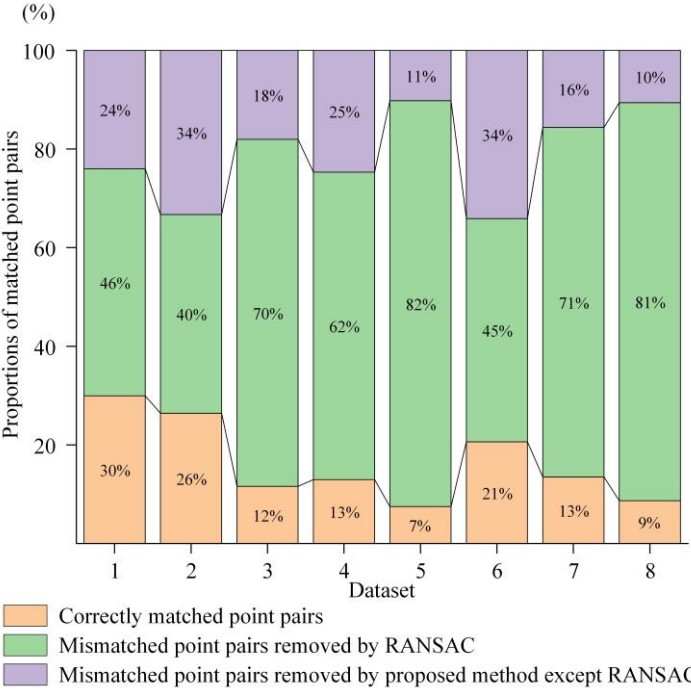

**Figure 5.** Proportions of matched point pairs in the UAV image used by the proposed method.

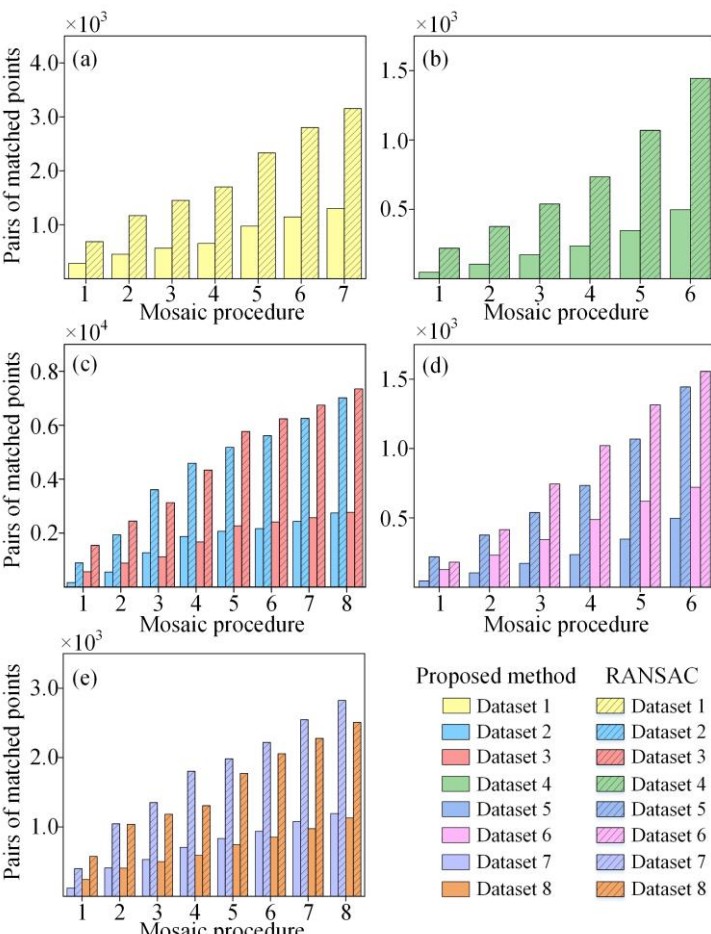

**Figure 6.** The UAV images used for crop growth monitoring using the proposed method and random sample consensus (RANSAC) to obtain the cumulative number of correctly matched feature points. (**a**) and (**b**) show the cumulative numbers of correctly matched feature points for the visible and thermal infrared UAV images of wheat, respectively; (**c**), (**d**), and (**e**) show the cumulative numbers of correctly matched feature points for the visible, thermal infrared, and near-infrared UAV images of rice, respectively.

### 4.3. Applicability of Algorithm to Images from Different Sources

Visible and near-infrared UAV images all contain abundant image information, and the image contrast is high [40,57]. The standard SIFT method extracts excess feature points, and the time consumption is extensive, which reduces the efficiency of UAV image mosaicking [58]. Due to the high contrast in visible and near-infrared UAV images, the proposed method maintains the number of feature points in an appropriate threshold range, improves the mosaicking efficiency, and ensures the mosaicking accuracy (Figure 7a,c).

The resolution and contrast of images acquired by the thermal imaging sensor are both relatively low [59]. The standard SIFT method conducts mosaicking on the thermal infrared images. The number of extracted feature points is fewer than needed, and obtaining the correct matching results is difficult, causing failure in the mosaicking [42]. By reducing $D_c$ to increase the number of matched feature points (Figure 7b), the proposed method produces sufficient feature points from thermal infrared images to calculate the transformation model and ensure the mosaicking accuracy. Therefore, this method effectively achieves thermal infrared image mosaicking and solves the problem of the standard SIFT method failing to mosaic such images.

In previous studies, visible, near-infrared, and thermal infrared UAV images all have applicable mosaicking methods and achieve relatively good results [41,60,61]. However, at most, these methods

are applicable to only two types of UAV images, and a universal mosaicking method for the three types of UAV images (i.e., visible, near-infrared, and thermal infrared) is lacking. The experiments in this study prove that the proposed method has high SSIM values for the mosaicking of visible, near-infrared, and thermal infrared UAV images. This result indicates that the proposed method can successfully conduct mosaicking on various types of UAV images used for crop growth monitoring and that the mosaicking accuracy is high. Thus, this method meets the needs identified in previous studies for a universal mosaicking method applicable to visible, near-infrared, and thermal infrared UAV images of farmland.

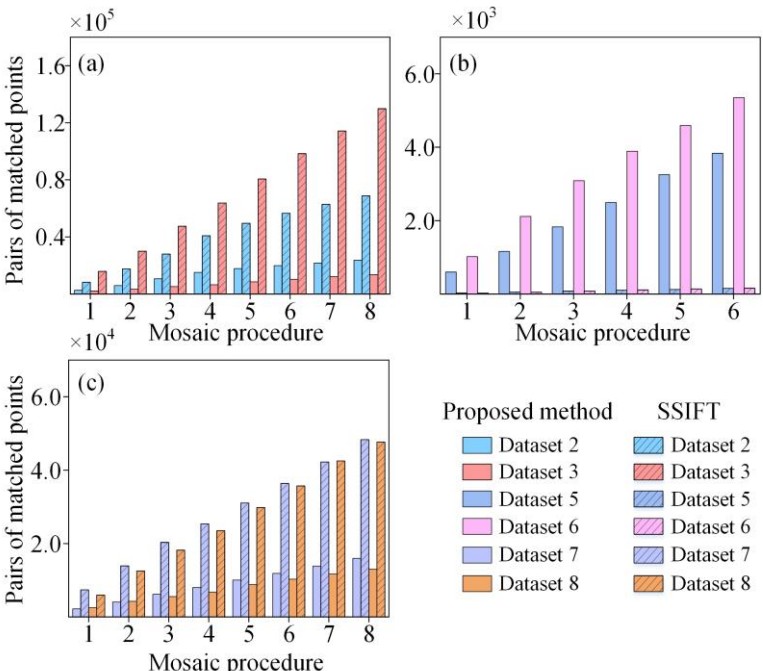

**Figure 7.** Cumulative numbers of feature points for rice UAV images processed using SSIFT and proposed method without deleting the mismatched points. (**a**), (**b**), and (**c**) are the cumulative numbers of matched feature points for the visible, thermal infrared, and near-infrared UAV images of rice, respectively.

*4.4. Applicability of Algorithm to Images from Different Growing Periods*

Rice has different morphological characteristics during the jointing and maturity stages. The mosaicking times for Dataset 2 and Dataset 3, Dataset 5 and Dataset 6, and Dataset 7 and Dataset 8, and the SSIM values indicate that the mosaicking times and accuracies of UAV images used for crop growth monitoring are not consistent during different growing stages. The physiological and morphological differences in crops and the environment during different growing stages affect the mosaicking process. During the jointing stage, the young panicles of rice begin to differentiate, and basal internodes begin to elongate. The plant-type characteristics of the community are mainly reflected by the various characteristics of leaves [57,58]. Although the stages of growth and development between individual plants are very different, UAV images taken at an altitude of 50 m cannot finely capture the differences between individuals. During the maturity stage, the grain filling of rice panicles is complete, the rice bends, and the plant volume is even larger than that during the jointing stage [62,63]. In an environment with adequate sunshine, the panicles produce dense shadows, and as the rice grows, its coverage in images gradually increases. These factors cause more complicated content and structure in UAV images during the maturity stage and affect the mosaicking results. In comparison with Dataset 2, Dataset 5, and Dataset 7, the numbers of feature point pairs extracted from Dataset 3, Dataset 6, and Dataset 8 are greater, and the numbers of deleted mismatched point pairs are also higher. Therefore, the UAV images during the maturation stage

have lower efficiency of matching feature points, and the mismatch situation is also more serious than those of the UAV images during the shooting period.

The method proposed in this paper can optimize the number of feature point pairs matched for a total of six image sets, e.g., Datasets 2, 3, 5, 6, 7, and 8 (Figure 7), and therefore effectively improve the mosaicking efficiency (Table 2). Moreover, the SSIM values for the mosaicking of the six groups of images are improved to different extents, which also indicates that the proposed method can improve the mosaicking accuracy of images during different growing stages and that the difference between the mosaicking accuracy of different images is very small (Table 2). The proposed method can be successfully applied to image mosaicking crop growth monitoring during different growing stages. However, in comparison with images acquired during the jointing stage, images acquired during the maturity stage used for crop growth monitoring take a longer time to mosaic.

### 4.5. The Use of Mosaics or Orthomosaics in Crop Growth Monitoring

In the domain of crop growth monitoring, some researchers generate orthomosaics rather than mosaics [64]. Orthomosaics pay more attention to spatial accuracy and are more suitable for geo-referenced analysis after data acquisition than mosaics [29]. However, most crop growth monitoring applications are used as real-time preliminary assessment tools. Furthermore, the generation of orthomosaics is time-consuming and cannot satisfy the demand of real-time crop growth monitoring. With the purpose of real-time applications, previous researchers found that mosaicking with a proper strategy is very efficient in crop growth monitoring [65,66]. Moreover, image stitching is recommended for a small field, when the terrain is sufficiently flat [67]. The applications of crop growth monitoring or phenotyping are usually conducted over a small area with a flat land surface [68,69]. Hence, the proposed method meets the above requirements and can improve the efficiency as a preliminary assessment tool by integrating the initial knowledge of the acquired crop growth monitoring images. Meanwhile, we have not yet tested the proposed method on orthomosaics, which should be studied in future work.

## 5. Conclusions

In this paper, we analyzed the characteristics of UAV images for crop growth monitoring and applied a mosaicking method using the SIFT algorithm. This method uses the image contrast to extract the contrast threshold in DOG scale-space that is most suitable for UAV images for crop growth monitoring. Furthermore, the proposed method was combined with the relative position relationship of images to delete mismatched point pairs to improve the time efficiency and spatial accuracy of mosaicking. The experimental results indicate that in comparison with the SSIFT and PS methods, the method proposed in this paper improves time efficiency by 30%. The results of the proposed method exhibit high similarity with the original UAV image, which can meet the needs of agricultural researchers to quickly and accurately monitor crop growth. The computational procedures are concise, and the applicability is strong. This method can be used for different types of UAV images for crop growth monitoring, thus providing a technical reference for UAV application for crop growth and phenotype monitoring.

**Author Contributions:** Conceptualization, X.Z.; Data curation, C.G.; Formal analysis, J.Z.; Funding acquisition, Y.Z.; Methodology, X.Z.; Project administration, W.C.; Software, J.Z.; Supervision, Y.T.; Visualization, X.Q.; Writing—original draft, J.Z.; Writing—review & editing, X.Z.

**Funding:** This research is supported by grants from the National Key R&D Program of China (No. 2016YFD0300607), the Fundamental Research Funds for the Central Universities (No. KYZ201810).

**Acknowledgments:** We thank Tao Cheng for his helpful suggestions on this manuscript.

**Conflicts of Interest:** The authors declare no conflict of interest.

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
