# Peer review of "Rapid Mosaicking of Unmanned Aerial Vehicle (UAV) Images for Crop Growth Monitoring Using the SIFT Algorithm"

_remotesensing, doi:10.3390/rs11101226_

Round 1

Reviewer 1 Report

The paper describes in correct and specif way. There are some references on similar problem. I suggest to improve the references with the work of Francesco Nex, Irene Aicardi, Filiberto Chiabrando, Markus Gerke, Marco Piras.

Author Response

May 14, 2019

Title: Rapid mosaicking of UAV images for crop growth monitoring using the SIFT algorithm

Dear Professor (reviewer #1)

We would like to thank you for giving us the constructive comment.

------------------------------------------------------------------------------------------------------------------------------------------

Point 1: The paper describes in correct and specific way. There are some references on similar problem. I suggest to improve the references with the work of Francesco Nex, Irene Aicardi, Filiberto Chiabrando, Markus Gerke, Marco Piras.

Response: We followed your suggestion and added 5 new references in lines 40, 101,  205 and 258 with their work to improve the references.

1. Piras M, Taddia G, Forno M G, et al. Detailed geological mapping in mountain areas using an unmanned aerial vehicle: application to the Rodoretto Valley, NW Italian Alps. Geomatics, Natural Hazards and Risk. 2017, 8(1), 137-149.

2. Remondino F, Barazzetti L, Nex F, et al. UAV photogrammetry for mapping and 3d modeling–current status and future perspectives. International archives of the photogrammetry, remote sensing and spatial information sciences. 2011, 38(1), C22.

3. Rottensteiner F, Sohn G, Gerke M, et al. ISPRS test project on urban classification and 3D building reconstruction. Commission III-Photogrammetric Computer Vision and Image Analysis, Working Group III/4-3D Scene Analysis. 2013, 1-17.

4. Chiabrando F, Donadio E, Rinaudo F. SfM for orthophoto to generation: A winning approach for cultural heritage knowledge. The International Archives of Photogrammetry, Remote Sensing and Spatial Information Sciences. 2015. 40(5), 91-98.

5. Irene A, Francesco N, Markus G, et al. An Image-Based Approach for the Co-Registration of Multi-Temporal UAV Image Datasets. Remote Sensing. 2016, 8(9), 779.

Reviewer 2 Report

In this paper a procedure for faster and more accurate mosaicking of UAV images is presented and discussed. The main idea is the dynamic determination of the image contrast threshold in the DOG scale-space when using the SIFT algorithm for the extraction of feature points from UAV images in crop growth monitoring.

The proposed methodology is based on a simple modification of the SIFT algorithm and, also, an improvement of the removal procedure of mismatched points using RANSAC is given. However, the results in mosaicking UAV images (images taken at low altitude and high degree of overlap) especially for the application of crop growth monitoring seem that have improved in terms of applicability (by using visible, near infrared and thermal infrared images), accuracy and time efficiency. The mathematical procedure of the proposed methodology is correct; the description of the methodology is adequate; and detail presentations and explanation of the results of the experiments are given. So, the paper can be accepted with only minor changes.

Some corrections are needed:

 -  The formula of the structural similarity index SSIM (equation 15) does not include the C3 factor, which is included on the structure (s) component of the formula: SSIM(x,y) = l(x,y) . c(x,y) . s(x,y)

 -    The numbering of the chapter titled as ‘Results and discussion’ (line 270) should  be corrected to ‘4’ (instead of ‘3’).

 -    In lines 512, 530 and 562 the sentence ‘Proceedings of the’ is mentioned two times (‘In Proceedings of the Proceedings of the …’)

Some additions are needed:

 -   The experiments are limited only to the examination of rice cultivation and only for 2 regions (fields). It is proposed that other crops should be examined too, that may exist in the region of the experiment or, even better, in some other region. Thus, the applicability of the proposed methodology and its reliability will be proved.

 -   The use of the structural similarity index (SSIM) instead of RMSE for the testing of the results using the proposed methodology it is sufficiently justified (chapter 3.2). However, for testing and comparing the accuracy of the results of mosaicking using the SSIFT, the proposed method and the PS, GCPs need to be measured; orthomosaics must be compiled; and the RMSE index must be used.

Author Response

May 14, 2019

Title: Rapid mosaicking of UAV images for crop growth monitoring using the SIFT algorithm

Dear Professor (reviewer #2)

We would like to thank you for giving us constructive suggestions.

------------------------------------------------------------------------------------------------------------------------------------------

Point 1: The proposed methodology is based on a simple modification of the SIFT algorithm and, also, an improvement of the removal procedure of mismatched points using RANSAC is given. However, the results in mosaicking UAV images (images taken at low altitude and high degree of overlap) especially for the application of crop growth monitoring seem that have improved in terms of applicability (by using visible, near infrared and thermal infrared images), accuracy and time efficiency. The mathematical procedure of the proposed methodology is correct; the description of the methodology is adequate; and detail presentations and explanation of the results of the experiments are given. So, the paper can be accepted with only minor changes.

Response: We appreciate that you are interested in our work, and thank you for your valuable and helpful comments.

Point 2: The experiments are limited only to the examination of rice cultivation and only for 2 regions (fields). It is proposed that other crops should be examined too, that may exist in the region of the experiment or, even better, in some other region. Thus, the applicability of the proposed methodology and its reliability will be proved.

Response: We followed your suggestion. We evaluated the proposed method on wheat growth monitoring at jointing stage in Region 1 and rice growth monitoring at jointing and maturity stages in Region 2 (Table 1 in line 251). Dataset 1 and Dataset 4 are wheat images while the other 6 datasets are rice images. Besides, we will share our codes of this method and wish others to evaluate the reliability of this method on other cultivars.

Point 3: The use of the structural similarity index (SSIM) instead of RMSE for the testing of the results using the proposed methodology it is sufficiently justified (chapter 3.2). However, for testing and comparing the accuracy of the results of mosaicking using the SSIFT, the proposed method and the PS, GCPs need to be measured; orthomosaics must be compiled; and the RMSE index must be used.

Response: Thanks for your suggestion. Although the general whole workflow of UAV applications contains stitching, orthorectifying and geo-referencing images [1], we just focus on stitching images for real-time crop growth monitoring. Moreover, the RMSE is always used to evaluate the accuracy of geo-referencing not the stitching. The accuracy of geo-referenced mosaics or orthomosaics can be evaluated by RMSE with ground control points (GCPs) [2], because the geo-locations of GCPs are measured and predicted. However, the geo-locations of GCPs in mosaics or orthomosaics without geo-referencing are not predicted, so we can’t apply RMSE as an accuracy evaluation index.

Besides, as a visual model-based indicator, SSIM is more in line with human visual characteristics and can be used, instead of RMSE or any subjective performance assessment methods, to evaluate performance of image stitching [3, 4].

1. Gómez-Candón D, De Castro A I, López-Granados F. Assessing the accuracy of mosaics from unmanned aerial vehicle (UAV) imagery for precision agriculture purposes in wheat Precision Agriculture. 2014. 15(1), 44-56.

2. Lin Y, Medioni G. Map-enhanced UAV image sequence registration and synchronization of multiple image sequences. 2007 IEEE Conference on Computer Vision and Pattern Recognition. 2007, 1-7.

3. Rojas J, Martinez C, Mondragon I, et al. Towards image mosaicking with aerial images for monitoring rice crops. Advances in Automation and Robotics Research in Latin America. 2017, 279-296.

4. Moussa A, El-Sheimy N. a fast approach for stitching of aerial images. International Archives of the Photogrammetry, Remote Sensing & Spatial Information Sciences. 2016, 41, 769-774.

Point 4: The formula of the structural similarity index SSIM (equation 15) does not include the C3 factor, which is included on the structure (s) component of the formula:

SSIM(x,y) = l(x,y) . c(x,y) . s(x,y)

Response: We followed your suggestion and removed the C3 factor.

Point 5: The numbering of the chapter titled as ‘Results and discussion’ (line 270) should  be corrected to ‘4’ (instead of ‘3’).

Response: We followed your suggestion and modified the chapter title.

Point 6: In lines 512, 530 and 562 the sentence ‘Proceedings of the’ is mentioned two times (‘In Proceedings of the Proceedings of the …’)

Response: We followed your suggestion and modified the sentence.

Reviewer 3 Report

The english must be revised. There are a few phrases that need to be rewritten, there are some small typo errors and some 

From lines 42 to 55. This text must be rewritten to make more sense. Different things are mixed: operational questions linked to UAV’s flight and processing questions. The text must be organised in a more logical way, clearly separating the different work steps.

line 57 to 60. Global Mapper can indeed mosaic images and can read LIDAR data but it cannot build a point cloud or a true orthomosaic from UAV imagery using 3D reconstruction. The differences between Photoscan or PIX4D and Global Mapper must be clarified.

The figure 2 shows a flight path and image overlap. It is recommended, in any UAV survey, that the images have, at least 60% overlap in both X and Y directions to have a meaningful and corrected orthomosaic in the end. The overlap shown in figure 2 will result in extensive areas of no overlap, with only one projection in potentially thousands of points. In my opinion. this draw must be completely remade. The authors, after, say that the imagery was processed in Photoscan. If the flight plan was done according to this draw, no alignment of photos would be obtained at all in Photoscan. 

Table 1 - Resolution must include the unit (pixels) or clarify that it’s the sensor matrix size. 

In results and discussion, to compare the processing methods, the authors must first clarify which was the precise flight planning and the amount of overlap. Many information is missing concerning this first operational part of the work. What software was used, what was the geometry of the flight, the amount of overlap, etc, etc. Photoscan, for instance, is very sensitive to these operational steps. A poor overlap will increase the processing times or will completely ruin the obtainment of final Digital Surface Models or Orthomosaics. 

Although the concept of the new proposed methodology seems to be interesting and mathematically well supported by the authors, the paper text has serious flaws in clarifying the flight parameters that where used to obtain the used imagery, the parameters that were used in Photoscan and other important issues that could be important to understand the comparison between processing methods. It is strongly recommended that the authors can rewrite an important part of the methodology and include more information. Probably it is recommended that more parameters used in Photoscan, screen captures of the results, the flight pattern used, portions of the Photoscan report and so on can be included in the text. 

Author Response

May 14, 2019

Title: Rapid mosaicking of UAV images for crop growth monitoring using the SIFT algorithm

Dear Professor (reviewer #3)

We would like to thank you for giving us constructive suggestions.

------------------------------------------------------------------------------------------------------------------------------------------

Point 1: From lines 42 to 55. This text must be rewritten to make more sense. Different things are mixed: operational questions linked to UAV’s flight and processing questions. The text must be organized in a more logical way, clearly separating the different work steps.

Response: Thanks for your suggestion. We followed your suggestion and separated the original part into two paragraphs from lines 39 to 56.

Point 2: line 57 to 60. Global Mapper can indeed mosaic images and can read LIDAR data but it cannot build a point cloud or a true orthomosaic from UAV imagery using 3D reconstruction. The differences between Photoscan or PIX4D and Global Mapper must be clarified.

Response: We followed your suggestion and added the description on the differences between Photoscan or PIX4D and Global Mapper from lines 57 to 67.

Point 3: The overlap shown in figure 2 will result in extensive areas of no overlap, with only one projection in potentially thousands of points. In my opinion. this draw must be completely remade. The authors, after, say that the imagery was processed in Photoscan. If the flight plan was done according to this draw, no alignment of photos would be obtained at all in Photoscan.

Response: We followed your suggestion. We remade the draw in lines 215 and describe longitudinal overlap and transverse overlap in the caption. The longitudinal overlap is 75% and the transverse overlap is 70%. Such overlap can ensure that Photoscan correctly processes images according to Agisoft PhotoScan User Manual.

Point 4: Table 1 - Resolution must include the unit (pixels) or clarify that it’s the sensor matrix size.

Response: We followed your suggestion, and added the unit of the resolution.

Point 5: In results and discussion, to compare the processing methods, the authors must first clarify which was the precise flight planning and the amount of overlap. Many information is missing concerning this first operational part of the work. What software was used, what was the geometry of the flight, the amount of overlap, etc. Photoscan, for instance, is very sensitive to these operational steps. A poor overlap will increase the processing times or will completely ruin the obtainment of final Digital Surface Models or Orthomosaics.

Response: Thanks for your suggestion. We followed your suggestion and added the details of flight plan in section 3.1 from lines 231 to 248.

Point 6: Although the concept of the new proposed methodology seems to be interesting and mathematically well supported by the authors, the paper text has serious flaws in clarifying the flight parameters that where used to obtain the used imagery, the parameters that were used in Photoscan and other important issues that could be important to understand the comparison between processing methods. It is strongly recommended that the authors can rewrote an important part of the methodology and include more information. Probably it is recommended that more parameters used in Photoscan, screen captures of the results, the flight pattern used, portions of the Photoscan report and so on can be included in the text.

Response: Thanks for your suggestion. We followed your suggestion and added more descriptions on the Photoscan reports with a new Table 3 in lines 304. Besides, combined with comment 5, the details of flight plan were added in section 3.1.

Reviewer 4 Report

Dear authors,

This manuscript describes a new method that allows the faster and more efficient UAV image mosaicking. To achieve this goal the method proposed uses existing image processing algorithms such as Difference of Gaussian scale-space and random sample consensus. Despite that this is a really interesting topic and useful in the field of image processing, I think the manuscript can be improved to be presented in a clearer way.

Introduction in my point of view is good enough requiring some minor spell checking. However, methodology should be clearer. The paper is not easy to read, in my point of view it is needed to start with the experiment and then talk about image processing. I think the improved removal of mismatched points section should be also clearer, maybe using a diagram it is possible to explain the process in a better way. It is not easy to understand that method just using words. Figure 1 resolution must be improved as well as captions in that figure. It would be better to use larger images.

There are some minor errors such as:

Line 38. Precision agriculture

Line 44. It is not possible to understand the sentence "continuous differ in angles of view differ"

Lines 171-174 don't have information

Figure 2. Caption inside the figure should say longitudinal overlap and transverse overlap

Table 1. Resolution unit is not written

Table 2. Dataset size unit is not written

Figure1. Sum of Dataset 2 and 5 is not 100% 

Line 652. there is a typo in reference 58

Author Response

May 15, 2019

Title: Rapid mosaicking of UAV images for crop growth monitoring using the SIFT algorithm

Dear Professor (reviewer #4)

We would like to thank you for giving us constructive suggestions.

------------------------------------------------------------------------------------------------------------------------------------

Point 1: The paper is not easy to read, in my point of view it is needed to start with the experiment and then talk about image processing.

Response: In order to clarify the structure of the paper, we added a new paragraph to describe the organization of the paper in the introduction section from lines 111 to 116.

Point 2: I think the improved removal of mismatched points section should be also clearer, maybe using a diagram it is possible to explain the process in a better way.

Response: We followed your suggestion and revised the section from lines 198 to 228. Besides, we redesigned the Figure 2 and added two sub-figures (Figure 2.b and Figure 2.c) to explain our approach more clearly.

Point 3: Figure 1 resolution must be improved as well as captions in that figure. It would be better to use larger images.

Response: We followed your suggestion and remade Figure 1 according to the Instructions for Authors of Remote sensing.

Point 4: Line 38. It is ‘Precision agriculture’ rather not ‘precise agriculture’. 

Response: Thanks for your suggestion, and we followed it.

Point 5: Line 44. It is not possible to understand the sentence "continuous differ in angles of view differ"

Response: We followed your suggestion and rewrote the section from lines 39 to 44.

Point 6: Lines 171-174 don't have information

Response: We followed your suggestion and deleted the blank lines.

Point 7: Figure 2. Caption inside the figure should say longitudinal overlap and transverse overlap. Figure5. Sum of Dataset 2 and 5 is not 100%

Response: We followed your suggestion and remade Figure 2. We described longitudinal overlap and transverse overlap in the caption of Figure 2. For Figure 5, we recalculated the values and modified the draw.

Point 8: Table 1. Resolution unit is not written. Table 2. Dataset size unit is not written

Response: We followed your suggestion and revised Table 1 and Table 2.

Point 9: Line 652. there is a typo in reference 58

Response: We corrected it.
